# Spiny-Cheek Crayfish, *Faxonius limosus* (Rafinesque, 1817), as an Alternative Food Source

**DOI:** 10.3390/ani11010059

**Published:** 2020-12-30

**Authors:** Natalia Śmietana, Remigiusz Panicz, Małgorzata Sobczak, Przemysław Śmietana, Arkadiusz Nędzarek

**Affiliations:** 1Department of Meat Sciences, Faculty of Food Sciences and Fisheries, West Pomeranian University of Technology in Szczecin, Kazimierza Królewicza Street 4, 71-550 Szczecin, Poland; natalia.smietana@zut.edu.pl (N.Ś.); malgorzata.sobczak@zut.edu.pl (M.S.); 2Institute of Marine and Environmental Sciences, University of Szczecin, Adama Mickiewicza Street 18, 70-383 Szczecin, Poland; przemyslaw.smietana@usz.edu.pl; 3Department of Aquatic Bioengineering and Aquaculture, Faculty of Food Sciences and Fisheries, West Pomeranian University of Technology in Szczecin, Kazimierza Królewicza Street 4, 71-550 Szczecin, Poland; arkadiusz.nedzarek@zut.edu.pl

**Keywords:** aquatic food, freshwater crayfish, meat colour, nutrient requirements, sensory analysis, structure, texture

## Abstract

**Simple Summary:**

Freshwater crayfish species are critical for both local communities and the modern food industry. The study characterised the meat of invasive spiny-cheek crayfish, analysing the yield, basic chemical composition, nutritional value, as well as culinary value. Crayfish meat has high nutritional parameters due to favourable fatty acid and amino acid profiles, as well as balanced mineral content. Crayfish meat is an alternative to livestock meat in the human diet and draws attention of the food industry to the processing of underutilised resources of high-quality aquatic species.

**Abstract:**

The aim of the study was to present a comprehensive characterisation of crayfish meat, which is crucial to assess its potential usefulness in the food industry. To this end, we assessed the yield, basic chemical composition (protein, fat, minerals), nutritional value (amino acid and fatty acid profiles, essential amino acid index (EAAI), chemical score of essential amino acids (CS), hypocholesterolaemic/hypercholesterolaemic ratio (h/H), atherogenicity (AI) and thrombogenicity (TI) indices), as well as culinary value (lab colour, texture, sensory characteristics, structure) of the meat of spiny-cheek crayfish (*Faxonius limosus*) (*n* = 226) from Lake Sominko (Poland) harvested in May–September 2017. Crayfish meat, especially that from the abdomen, was shown to have high nutritional parameters. It is lean (0.26% of fat), with a favourable fatty acid profile and a very high quality of fat (PUFA (sum of polyunsaturated fatty acids):SFA (sum of saturated fatty acids), n-6/n-3, h/H, AI, TI) and protein (high CS and EAAI). It is also a better source of Ca, K, Mg, Na, P, and Cu than meat from slaughter animals. Hence, crayfish meat can be an alternative to livestock meat in the human diet. Owing to its culinary value (delicateness, weak game flavour, and odour), it meets the requirements of the most demanding consumers, i.e., children and older people.

## 1. Introduction

In recent years, there has been an increase in interest in raw materials used to create new products to meet the growing demand of customers. The meat of freshwater crayfish is included in this category. It is consumed in various countries of Asia, North America, and Europe. In Europe, crayfish were found on tables from the Middle Ages to the 1920s. Later, the availability and popularity of crayfish for consumption decreased, e.g., owing to a rapid decline in the populations of local species caused by the emergence of crayfish plague in the second half of the 19th century. Hence, consumers started to consider crayfish as a very rare animal [1]. The second reason for the decrease in the population of European crayfish species was the introduction of spiny-cheek crayfish (*Orconectes* [*Faxonius*] *limosus* [Rafinesque, 1817]), originating in North America, and signal crayfish (*Pacifastacus leniusculus*) into water reservoirs at the end of the 20th century [2]. The two species, compared with native species, have superior reproductive performance, tolerance to environmental conditions, and resistance to diseases [2]. Owing to these characteristics, the species have spread in the natural environment and outnumbered indigenous species [2]. Spiny-cheek crayfish are currently widespread in multiple European countries [3,4]. Their presence has been recorded in Germany, Poland, the Czech Republic, Netherlands, Luxembourg, France, United Kingdom, Russia, Slovakia, Romania, Serbia, Croatia, Switzerland, Austria, Belgium, Spain, Belarus, Latvia, Lithuania, Hungary, and Italy [3], and since 2008 in Estonia [4]. In all those countries, spiny-cheek crayfish poses a threat to the indigenous crayfish species. The population size of spiny-cheek crayfish is growing steadily and depends on the location and type of water reservoir. In Poland, 57 sites of occurrence of the species were recorded in 1959 compared with 1383 sites recorded in 2004 [5]. In Nottinghamshire (UK), spiny-cheek crayfish colonised 16 ha of Attenborough Lake within 4 to 5 years [6]. In the German Lake Grosser Vätersee, the species accounts for up to 49% of the biomass of macroinvertebrates, which is equivalent to 35% and 81% of the biomass of herbivorous and predatory fish, respectively [7]. In Lake Varese, the population of this species is estimated at 6.4–16.7 individuals m^−2^ of the reservoir, depending on the site [8].

Rebuilding the populations of indigenous crayfish could be aided by harvesting invasive species that may be a source of edible and non-edible raw material [9,10,11,12]. The effectiveness (yield) of these catches would be affected by the type of water reservoir, season of the year, as well as method and gear used for the catches [8,13]. However, use of spiny-cheek crayfish in the food industry would only be possible once studies assessing the yield and properties of crayfish meat have been conducted. The available literature includes studies of the nutritional value of the meat of different crayfish species (*Procambarus clarkii, Astacus leptodactylus, Astacus astacus, Cherax quadricarinatus*) and use of their inedible parts (exoskeleton) and edible parts (meat and meat protein preparations) in the food industry [10,14,15]. Crayfish meat has a high nutritional value, i.e., high protein content (12.9–17.8%) with a relatively low fat content (0.14–1.69%) [16,17,18] and a desired fatty acid composition and low cholesterol content (75.9–81.2 mg g^−1^) [17]. However, there are no detailed data comprehensively describing the quality and nutritional value of the meat of spiny-cheek crayfish. Studies by Stanek et al. [19,20,21] only concerned the characteristics of the abdomen meat fat of this crustacean species. This information is insufficient to estimate the usefulness of this raw material in the food industry. Therefore, the aim of this study was to assess the nutritional value (chemical composition, amino acid and fatty acid profiles, element content) and quality parameters (pH, colour, texture, and sensory properties) of the meat of spiny-cheek crayfish.

## 2. Materials and Methods

### 2.1. Crayfish Sampling and Basic Characteristics

Live whole spiny-cheek crayfish individuals (*n* = 226) were caught by free diving from Lake Sominko (Poland), (54°4′47″ N 17°52′48″ E) in May–September of 2017. Immediately after sampling. the crayfish were placed in a tank filled with water and transported to the laboratory of the Department of Meat Sciences (ZUT in Szczecin, Poland). Upon arrival, individuals were kept at 4 °C overnight and subsequently euthanised and weighed using an electronic balance (PS6100.R2.M, Radwag, Radom, Poland). Individuals were dissected, and the abdomen and a chela, including the meat and hard parts thereof, were weighed. The yields (%) of the meat and hard parts were calculated using the following formulae:(1)$$\mathrm{Yield}\;\mathrm{of}\;\mathrm{hard}\;\mathrm{parts}\;\left(\%\;\mathrm{of}\;\mathrm{body}\;\mathrm{part}\right)=\frac{\mathrm{weight}\;\mathrm{of}\;\mathrm{hard}\;\mathrm{parts}}{\mathrm{weight}\;\mathrm{of}\;\mathrm{body}\;\mathrm{parts}}\;\times \;100\%$$
(2)$$\mathrm{Yield}\;\mathrm{of}\;\mathrm{meat}\;\left(\%\;\mathrm{of}\;\mathrm{body}\;\mathrm{part}\right)=\frac{\mathrm{weight}\;\mathrm{of}\;\mathrm{meat}}{\mathrm{weight}\;\mathrm{of}\;\mathrm{body}\;\mathrm{parts}}\;\times \;100\%$$
(3)$$\mathrm{Yield}\;\mathrm{of}\;\mathrm{meat}\;\left(\%\;\mathrm{of}\;\mathrm{body}\right)=\frac{\mathrm{weight}\;\mathrm{of}\;\mathrm{meat}}{\mathrm{weight}\;\mathrm{of}\;\mathrm{body}}\;\times \;100\%$$

Ethics approval was obtained from the ethics committee of the Faculty of Food Sciences and Fisheries (ZUT in Szczecin, Poland, number 517-08-026-7724/17). We adhered to the “Guidelines for the treatment of animals in behavioural research and teaching” published in Animal Behaviour [22].

### 2.2. Chemical Analyses

The chemical composition of minced abdomen and chelae meat (*n* = 32) was determined according to AOAC (Association of Official Analytical Chemists) procedures [23]. Moisture was obtained after drying samples in an oven at 105 °C for 24 h, while ash content was determined after incineration at 550 °C for 6 h. To assess the level of crude protein in crayfish meat, the content of N was first assessed by digesting 0.500 ± 0.01 g of meat samples in the mixture of 95% sulfuric acid (Chempur, Piekary Śląskie, Poland) and 30% hydrogen peroxide (Chempur, Piekary Śląskie, Poland) in equal volumes of 10 mL each, together with a catalyst—Kjeldahl tablets (MERCK KGaA, Darmstadt, Germany). A digestion step was performed using Heating Digester DK6 and DK8 (VELP Scientifica, Usmate Velate, Italy) following the thermal profile of 30 min at 180 °C, 30 min at 280 °C, and 30 min at 380 °C. Subsequently, crude protein was measured by determining the nitrogen content (N × 6.25) according to the Kjeldahl method, using a Tecator Kjeltec 2100 distillation unit (FOSS Analytical Co., Ltd., Jiangsu, China). Crude lipid was determined gravimetrically, after Soxhlet lipid extraction on a Tecator Soxtec System HT 1043 (FOSS Analytical Co., Ltd., Jiangsu, China). The procedure of lipid extraction was performed according to the default settings recommended by the equipment vendor, i.e., at 90 °C in the presence of petroleum ether 40/60 pure p.a. (Chempur, Piekary Śląskie, Poland). The fatty acid profiles in abdomen meat samples (*n* = 50) were quantified using gas chromatography (GC) with a flame ionisation detector (FID). Briefly, fatty acids were determined as fatty acid methyl esters (FAME), and individual FAME were identified by comparing their retention times with those of pure standards. Analyses were carried out in triplicate on an Agilent 6890N Network Gas Chromatograph (Agilent Technologies; Palo Alto, CA, USA) equipped with a 7683 automatic liquid sampler and flame ionisation detectors. The amino acid profile of proteins (*n* = 20) in the abdomen and chelae meat samples was determined by High Performance Liquid Chromatography (HPLC) using an AAA 400 amino acid analyser (Ingos, Prague, Czech Republic). The chromatograms were analysed using the CHROMuLAN V 0.88 program (PiKRON, Prague, Czech Republic) by comparison with the standard chromatogram, taking into account dilution and weight. All analyses were performed in triplicate. Energy value was calculated using the relative percentage of each nutrient (protein and fat) which was multiplied by the correction factors, 4 kcal g^−1^ (17 kJ g^−1^) and 9 kcal g^−1^ (37 kJ g^−1^) for protein and fat, respectively, as described in Regulation (EU) No. 1169/2011 [24].

Protein quality was described by the chemical score (CS) of essential amino acids (EAA) and the essential amino acids index (EAAI). The CS was calculated in relation to a reference scoring pattern suggested by FAO/WHO/UNU [25] according to the following equation:(4)$$\mathrm{CS}\;=\;\frac{g\;\mathrm{EAA}\;\mathrm{in}\;\mathrm{tested}\;\mathrm{protein}}{g\;\mathrm{EAA}\;\mathrm{in}\;\mathrm{pattern}\;\mathrm{protein}}\;\times \;100$$

The essential amino acids index (EAAI) was calculated according to the equation described by Shahidi and Synowiecki [26]:(5)$$\mathrm{EAAI}\;=\;100\;\times \;\sqrt[{n}]{\frac{a}{a_{p}}\times \;\frac{b}{b_{p}}\times \ldots \times \;\frac{i}{i_{p}}}$$
where a, b,…, i—content of histidine, isoleucine, leucine, lysine, SAA (sulphur amino acids—sum of methionine and cysteine), AAA (aromatic amino acids—sum of phenylalanine, tyrosine, and tryptophan), threonine and valine in sample, a_p_, b_p_,…, i_p_—content of histidine, isoleucine, leucine, lysine, SAA (sulphur amino acids—sum of methionine and cysteine), AAA (aromatic amino acids—sum of phenylalanine, tyrosine, and tryptophan), threonine and valine in protein standard [25], n—number of amino acids.

Fat quality was described by the following factors: SFA (sum of saturated fatty acids), MUFA (sum of monounsaturated fatty acids), PUFA (sum of polyunsaturated fatty acids), h/H (hypocholesterolaemic/hypercholesterolaemic ratio), AI (index of atherogenicity), and TI (index of thrombogenicity). These factors were calculated using the fallowing equations [27,28,29]:SFA = (C12:0 + C14:0 + C15:0 + C16:0 + C17:0 + C18:0 + C20:0 + C21:0 + C22:0)(6)
MUFA = (C16:1n7 + C17:1n7 + C18:1n9t + C18:1n9c + C20:1n5 + C20:1n9)(7)
PUFA = (C18:2n6t + C18:2n6c + C18:3n3 + C20:2n6 + C20:3n3 + C20:3n6 + C20:4n6 + C20:5n3 + C22:6n3)(8)
h/H = Σ(C18:1n 9, C18:1n 7, C18:2n 6, C18:3n 6, C18:3n 3, C20:3n 6,C20:4n 6, C20:5n 3, C22:4n 6, C22:5n 3, C22:6n 3)/Σ(C14:0, C16:0)(9)
AI = (C12:0 + 4 × C14:0 + C16:0)/((n-6)PUFA + (n-3)PUFA + MUFA)(10)
TI = (C14:0 + C16:0 + C18:0)/(0.5 × MUFA + 0.5 × (n-6)PUFA + 3.0 × (n-3)PUFA + (n-3)PUFA/(n-6)PUFA)(11)

### 2.3. Elemental Analysis

The abdomen and chelae meat samples (*n* = 24, each) of spiny-cheek crayfish were dissolved as described by Mistri et al. [30]. Samples of 0.8 ± 0.1 g (wet weight) were digested in 10 mL of concentrated ultrapure HNO_3_ (Merck, Darmstadt, Germany) in a Speedwave Xpert high-pressure microwave mineraliser (Berghof, Eningen, Germany). The DAK-100 reaction vessels used were made of TFM^TM^-PTFE (second generation of polytetrafluorethylene), and the digestion conditions were as follows: power—2000 W, hold time—25 min, and temperature—200 °C. After cooling, samples were transferred into volumetric flasks (25 mL) and diluted to the mark with deionised water (18.2 MΩ). Element measurements in crayfish tissues were carried out with a Hitachi ZA3000 Series Polarised Zeeman Atomic Absorption Spectrometer (Hitachi High-Technologies Corporation, Tokyo, Japan) equipped with a Zeeman background correction system. Ca, K, Na, and Mg were measured using flame atomic absorption spectroscopy (FAAS) in an air–acetylene flame. Al, Cd, Cu, Fe, Pb, and Zn were measured using graphite furnace atomic absorption spectroscopy (GFAAS). Radiation sources for elements were hollow-cathode lamps (HCL, Hitachi High-Technologies Corporation, Tokyo, Japan) at the appropriate wavelengths (in nm): 309.3 (for Al); 422.7 (for Ca); 228.8 (for Cd); 324.8 (for Cu); 248.3 (for Fe); 766.5 (for K); 285.2 (for Mg); 589.0 (for Na); 283.3 (for Pb); 213.9 (for Zn). The following matrix modifiers were also used: palladium (0.2% Pd in 5% HNO_3_, SIGMATIK, Wroclaw, Poland); Mg(NO_3_)_2_ and NH_4_H_2_PO_4_ (both 1000 mg L^−1^, Merck, Darmstadt, Germany), along with the caesium chloride–lanthanum chloride buffer solution acc. to Schinkel for atomic absorption spectroscopy (10 g L^−1^ CsCl and 100 g L^−1^ La, MERCK, Darmstadt, Germany). Phosphorus (P) was quantified colorimetrically in the same sample solutions as the other elements. The employed method, described by Jastrzębska [31], used ammonium molybdate and ascorbic acid as reducers, forming molybdenum blue. Absorbance at λ = 882 nm was measured on a UV-VIS spectrophotometer Spectroquant Pharo 300 (Merck, Darmstadt, Germany).

Calibration curves were established using certified standard solutions (1000 mg L^−1^) from Scharlau (Barcelona, Spain) for Mg, Ca, K, Na, and Fe, and from Merck (Darmstadt, Germany) for Al, Cd, Cu, P, Pb, and Zn. The range of standard concentrations included only the linear character of the calibration curve (the R-factor, determined by the spectrophotometer program, was always >0.9955). Limits of Quantification (LOQ) were as follows: 0.3 mg kg^−1^ (for Ca, K); 0.03 mg kg^−1^ (for Mg); 0.15 mg kg^−1^ (for Na); 1.0 µg kg^−1^ (for Al, Cd, Pb); 1.5 µg kg^−1^ (for Cu, Fe); 0.2 µg kg^−1^ (for Zn). The obtained results were assessed for accuracy and precision using certified reference material of Fish Muscle ERM BB422 (European Reference Materials, European Commission—Joint Research Centre, Institute for Reference Materials and Measurements, Geel, Belgium). The recovery of elements was within 95–105% and the precision for reference material was 1.3–11.4%. See Appendix A for comprehensive data analysis.

### 2.4. Assessment of Muscle Structure

Samples (5 × 5 × 5 mm) of raw chelae and abdomen meat (*n* = 20) were fixed for 12 h in Sannomiya solution, dehydrated using alcohol, and saturated in intermediate solutions (benzene, benzene: paraffin). Then, samples were embedded in paraffin blocks, trimmed, sectioned (10 ± 1 μm, Rotary Microtome MPS-2, Opta-Tech, Warsaw, Poland), stained with haematoxylin and eosin, and mounted on slides with Canadian lotion [32]. For each sample, three pieces of specimen were prepared, randomly selected, and examined by two members of the laboratory using an Eclipse E600 microscope (Nikon, Nikon, Japan) with a 100× objective. Specimens were screened for fibre cross-section area (CSA), fibre girth, horizontal (H) and vertical (V) diameter of fibre, and thickness of endomysium using the ROI (region of interest) tool in the NIS-Elements Basic Research software (Nikon Instruments Europe B.V, Warsaw, Poland), (Figure 1). Additionally, fibre shape was calculated as the H:V diameter ratio.

### 2.5. pH and Colour

pH was measured in quadruplicates for each sample (*n* = 20) of raw meat of chelae and abdomen. The measurement was done using a portable pH meter (CP-411, Elmetron, Zabrze, Poland) with a glass penetrating electrode. Before the analysis, the pH meter was calibrated using standard phosphate buffers (pH 4.00 and 7.00). Between measurements, the electrode was rinsed thoroughly with distilled water. The colour of raw meat of chelae and abdomen samples (*n* = 20) was assessed using a NR 20XE Precision Colorimeter (Shenzhen 3NH Technology Co., Ltd., Shenzhen, China) with φ20 mm extended aperture. L* (lightness), a* (redness), and b* (yellowness) were obtained automatically after a light shot had been discharged perpendicularly to the surface of abdomen and chelae meat. Measurements were done in triplicates. Whiteness index (WI) and chromaticity (C) were calculated using following equations:WI = 100 − [(100 − L)^2^ + a^2^+ b^2^]^0.5^(12)
C = (a^2^ + b^2^)^0.5^(13)

### 2.6. Texture

The texture of abdomen meat boiled for 4 min (*n* = 20) was measured in quintuplicate with Instron 1140 (Stable Instron, Bucks, UK) using a double compression test [33]. Briefly, a 120 mm^2^ plate compressed the sample twice to 80% of their original height, and parameters, such as hardness (N), cohesiveness (-), springiness (cm), and chewiness (N × cm), were measured. The crosshead speed was 50 mm min^−1^.

### 2.7. Sensory Analysis

Sensory evaluation of abdomen meat boiled for 4 min (*n* = 20) was conducted by a trained team, composed of four members [34]. Texture characteristics (springiness, cohesiveness, hardness, tenderness, moisture, juiciness, perceptibility of connective tissue, chewiness, fattiness, astringency), intensity of odour, and taste descriptors were evaluated. Intensity of these features was rated using a 5-point scale, where 1 point corresponded to the lowest and 5 points to the highest intensity.

### 2.8. Statistical Analysis

Data were analysed using STATISTICA for Windows (version 13.1, Krakow, Poland). The data were subjected to a two-way analysis of variance (ANOVA) and Tukey’s test to compare sample means. The significance level for Tukey’s test was 0.05.

## 3. Results and Discussion

### 3.1. Yield of Spiny-Cheek Crayfish Meat

Crayfish meat is considered to be a particular type of raw material due to its sensory properties, method of preparation, availability, and origin. Although it is currently not widely used for food purposes, it can become an alternative source of animal raw material in the near future. However, full use of crayfish meat by the food sector will only be possible once it has been thoroughly characterised. Our study showed that the percentage yield of meat in the abdomen and chelae in relation to the total body mass of crayfish were 8.67% and 2.69%, respectively (Table 1).

A higher yield of abdomen meat compared with that of the chelae was also demonstrated by a study by Berber and Balık [35] in *A. leptodactylus*, but in both cases, yield was higher (12.98% and 3.47%, respectively) than in our study. In turn, Thompson et al. [18] showed a more than doubled (24%) percentage yield of abdomen meat for red chelae crayfish, *Cherax quadricarinatus*, stocked into earthen ponds. The percentage of muscle and hard parts varies depending on the species, population, age, and degree of exoskeleton mineralisation. For example, *P. leniusculus* has a much more mineralised exoskeleton compared with *A. leptodactylus*. Consequently, with a comparable share of abdomen muscle (13.7% vs. 12.6%), exoskeleton mineralisation has a significant impact on the actual yields of abdomen meat in both species (7.5 g vs. 3.3 g) [36]. The degree of mineralisation of individual body parts of crayfish is a result of the function played by the body part, which is clearly demonstrated by the significantly higher (*p* ≤ 0.05) share of shell in chelae than in the abdomen reported in the study [37]. The thick armour of chelipeds and chelae allows crayfish to function more effectively in the environment; however, it complicates meat recovery from that body part. Appropriate technology is necessary for this purpose, as manual meat removal from chelae is usually economically inviable and involves a microbiological risk. For example, the industrial extraction of meat from chelae uses two-phase separation preceded by the crushing of chelae [38].

### 3.2. Nutritional Value of Spiny-Cheek Crayfish Meat

Chemical analysis did not reveal any significant differences (*p* > 0.05) in the content of protein, fat, ash, dry matter, and energy between abdomen and chelae meat (Table 2). Protein content in the meat of the abdomen (18.23 ± 0.732) and chelae (18.83 ± 1.476) of *F. limosus* was comparable to that in other crayfish species [19,39], marine and freshwater fish [40], and livestock [41,42]. Based on the results of this study and studies by other authors [19,21,43], the fat content in the meat of spiny-cheek crayfish is 0.26% and 0.24–1.35%, respectively, which allows considering this meat as lean. A similar amount of fat is found in the meat of lean fish (<2%) and in that of other crustaceans [43,44]. For example, for red swamp crayfish (*Procambarus clarkii*), El-Kholie et al. [45] reported approximately 1.99% of crude fat. Due to the low fat content, the energy value of the abdomen and chelae meat of spiny-cheek crayfish is low: 75.27 kcal 100 g^−1^ (319.59 kJ 100 g^−1^) and 77.52 kcal 100 g^−1^ (329.14 kJ 100 g^−1^), respectively. In livestock meat, the average fat content is 5–32% for pigs, 3.3–7.6% for cattle, 2.7–12% for sheep, and 1–9% for chickens [42,46]. In game, the values are 2.82%, 1.6%, 1.9%, and 1.3% for wild boar, roe deer, red deer, and moose, respectively [47]. Moreover, the energy value of livestock meat is higher than that of crayfish meat. For example, the energy value per 100 g is 115–146 kcal or 455–716 kJ for cattle, 134–353 kcal or 461–784 kJ for pig, 122 kcal or 473–767 kJ for sheep [41,42,48], 109–178 kcal for chicken, 106–136 kcal for turkey, and approximately 130 kcal for duck [41]. Based on the above list, it appears that in terms of fat and calorie content, the meat of spiny-cheek crayfish may be an alternative to livestock meat for consumers.

Analysis of the obtained data showed significant differences in the mineral content of the abdomen and chelae meat of spiny-cheek crayfish. The muscle of the abdomen had a significantly higher content of K, P (*p* ≤ 0.01) and Cd, Pb (*p* ≤ 0.05), and it had a significantly lower amount of Ca, Mg, Na, Fe, and Zn (*p* ≤ 0.01) as well as Al and Cu (*p* ≤ 0.05) than that of chelae (Table 3). Difference in the observed contents of the elements reflects the composition of the diet but also the function of muscles located in the abdomen and chelae [49,50,51]. When comparing the mineral content of the meat of spiny-cheek crayfish with that of other animal species, it can be demonstrated that a similar Ca, K, and Zn content is present in the meat of Chinese mitten crab (*Eriocheir sinensis*) [52]. Lower amounts of Ca, K, Mg, Na, P, and Cu and higher amounts of Fe and Zn have been recorded in the meat of cattle, sheep, pig [42,48], and chicken [53]. According to EU health claims legislation [24], beef, lamb, and pork can be classified as rich sources of several nutrients. Beef is a rich source of Zn and a source of Fe, K, and P; lamb is a rich source of Zn and a source of K and P; whereas pork is a source of Fe, Se, K, and P [42]. Our study showed that the meat of spiny-cheek crayfish is an even better source of Ca, K, Mg, Na, P, and Cu than livestock meat. The levels of Pb and Cd in the meat of the abdomen and chelae did not exceed the established food safety level of 0.50 μg g^−1^. The higher concentration of Cd and Pb in the meat of the abdomen compared with that of chelae might be explained by different biological and natural factors, as shown for green crab (*Carcinus maenas*). The accumulation rate depends on ion concentrations, hydration level, and also the volume of tissues, which can be interpreted as condition [54,55]. In our case, the level of dry matter (hydration) was similar for the investigated meat samples, but abdomen meat with its higher yield presumably had a higher capacity to accumulate heavy metals. Our study showed that the abdomen and chelae meat of spiny-cheek crayfish is toxicologically safe and can be a good source of macro- and microelements in human nutrition. The consumption of 100 g of chelae meat covers 103.7%, 54.8%, and 45.1% of the consumer’s daily demand for Zn, Ca, and P, respectively. The consumption of 100 g of abdomen meat mostly covers the demand for P (49.93%), Zn (17.08%), and K (11.75%) (Table 3). In comparison, the consumption of 100 g of beef covers 9%, 22%, and 30% of the demand for K, P, and Zn, respectively [48].

The nutritional value of raw materials and food products, in addition to the basic nutrients and minerals, is also determined by the nutritional value of protein based on its amino acid composition. Our study showed that the meat of the abdomen and chelae has a comparable amino acid composition. The exceptions were the significantly (*p* ≤ 0.01) higher levels of arginine and glycine in abdomen muscles and of alanine and proline in chelae muscles (Table 4).

The identified differences are likely to be directly associated with the function of these muscle groups, and in particular, with the activity of their metabolic pathways. For example, the higher level of arginine in abdomen muscles compared with that in chelae is associated with the functioning of the highly efficient arginine phosphate/arginine kinase system, which is crucial for burst locomotion, swimming, and egg incubation in females [64]. The higher level of alanine in chelae muscles in postmolt animals may be related to some aspects of electrolyte balance during the dilution of electrolytes associated with ecdysis [65]. Arg, Leu, and Lys were the most frequent amino acids in the EAA group, and Glu and Asp were the most frequent amino acids in the NEAA group. Comparison of amino acid profiles between the meat of *F. limosus* and livestock [41,66,67,68] and fish [69,70,71] showed that raw material collected from spiny-cheek crayfish has similar or higher free amino acid concentrations (Arg, Ile, Leu, Trp, Glu) than that from vertebrates. According to Claybrook [72], this is a manifestation of differing osmoregulatory needs. The essential amino acids to nonessential amino acids ratio (EAA:NEAA) in crayfish meat is at 1.14–1.15 and is higher than in the meat of chicken (0.94–1.02) [73] and duck (0.63) [66], but it is similar or slightly lower than in the meat of cattle (1.02–1.31) [41,68], pork (1.38), lamb (1.30) [41], and nutria (1.15–1.3) [74]. The high protein quality of abdomen and chelae meat was confirmed by chemical scores (CS) that, except for valine in chelae meat, were higher than 100 (Table 5).

Our study showed that the meat of spiny-cheek crayfish had a better nutritional protein quality in comparison with the FAO/WHO/UNU standards [25]. By analysing and recalculating the results obtained by other authors, it can be noted that the meat of cattle, pig, sheep [75], and rabbit [76] has higher CS values for His, Ile, Lou, Thr, and Val, similar CS values for Lys and Phe+Tyr, and lower CS values for Tyr compared with the meat of spiny-cheek crayfish. Moreover, the essential amino acid index (EAAI) calculated for the abdomen and chelae meat of *F. limosus* was 151.53% and 145.94%, respectively, and it was higher than that for the reference standard protein (Table 4). The EAAI values in this study are higher comparing with the meat of several species, e.g., 114% in lagoon crab *Calinectes latimanus* [77], 89% in innards *N. maculatus* [78], 50.4–82.9% in chicken [75,79], 80–81% in beef, pork, and mutton [75], and 128–136% in freshwater fish [80], similar as in rabbit meat—153% [76], and lower than that in sea bass (*Dicentrarchus labrax*) fillets—266% [81]. The level of protein, but also the CS and EAAI in the abdomen and chelae, showed that *F. limosus* meat is highly digestible and has a well-balanced amino acid composition.

The high nutritional value of crayfish meat protein can contribute to the wider use of this raw material for food purposes. However, it seems that abdomen meat is of greater importance than that from chelae. Due to its very low yield, chelae meat is not an economically viable raw material. However, it may be used in the production of, e.g., protein preparations, which are subsequently used in food processing [15,38]. We conducted a detailed analysis of the quality of the abdomen meat of spiny-cheek crayfish, which included assessment of the fatty acid (FA) profile, pH, colour, structure, texture, and sensory properties.

The FA profile analysis (Table 6) showed that approximately 44.4% of FA in abdomen meat are polyunsaturated fatty acids (PUFA), while saturated (SFA) and monounsaturated (MUFA) fatty acids constitute 28.8% and 26.8%, respectively. Therefore, it can be proposed that the proportion of PUFA, MUFA, and SFA in crayfish meat is more favourable than in pork [82,83], beef [84,85,86], lamb [87], chicken [53,87], and goose [88], all of which contain more SFA and MUFA than PUFA.

The higher value of the crayfish-derived raw material was confirmed by the PUFA:SFA ratio (1.45) which, according to nutritional recommendations, should be greater than 0.45 [89]. In comparison, the PUFA:SFA ratio was approximately 0.3 in pork [82,83], 0.25–0.79 in beef [85,86], 0.19–0.2 in lamb [90], and 0.6 in chicken [87]. In crayfish meat, palmitic acid (C16:0) was the most abundant SFA, the sum of oleic (C18:1n9c) and elaidic (C18:1n9t) acids was the most abundant MUFA, and eicosapentaenoic acid (EPA, C20:5n3) was the most abundant PUFA. In the PUFA subset, the quantities of n-3, n-6, and n-9 acids were similar (0.0850–0.0895 g 100g^−1^ of meat), and the n-3/n-6 ratio was 1.08 (n-6/n-3 = 0.93). The n-3/n-6 ratio was comparable with the results reported by other authors (0.72–1.06) for various crayfish species [19,20,90]. The obtained n-3/n-6 (or n-6/n-3) ratio for crayfish meat meets nutritional recommendations (>0.25 or <4.0) [89], and it is more beneficial than that calculated for the meat of several livestock species, e.g., 0.10–0.21 in pork [82,83], 0.08–0.18 in beef [85,86], 0.07 in rabbit [91], 0.07–0.12 in chicken [53,87], and 0.13–0.18 in goose [88]. Only for the meat of marine fish (e.g., Pacific herring, *Clupea harengus pallasi*; Pacific hake, *Merluccius productus*; sardine, *Sardinops sagax*; walleye pollock, *Theragra chalcogramma*), this ratio was much higher than that for crayfish meat and ranged between 7.35 and 18.66 [92]. In our study, 100 g of crayfish meat contained 0.0688 g of the sum of EPA and DHA (docosahexaenoic acid), which is much more than pork—0.0118 [83] and chicken—0.037 [87].

Additionally, h/H, AI, and TI calculated based on the FA profile of abdomen meat were 3.30, 0.29, and 0.29, respectively (Table 6), indicating a high nutritional value of crayfish meat fat. The h/H index indicates the influence of specific fatty acids on cholesterol metabolism, and generally the higher the value, the better. The value of the h/H index obtained in our study was higher than that reported for the meat of several livestock species, e.g., 1.8–2.66 in chicken [53,93], 2.6–2.8 in goose [88], 1.8 in beef [94], and 2.4 in pork [95]. Further comparisons of the h/H indices showed that the value calculated for *F. limosus* meat fell within the range for fillets of marine fish—3.1 [96] and freshwater common carp—3.4 [97], and it was lower compared with 5.9 in crab edible tissue [98]. In terms of human health, the AI and TI take into account the different effects that a single FA might have on human health, and lower values of these indices (<1.0 for AI, <0.5 for TI) in the diet are strongly recommended [96]. Our study showed that the abdomen meat of spiny-cheek crayfish meets these requirements and can be recommended for delaying atherosclerosis and thus for minimising the risk of cardiovascular disorders [28,29]. The AI and TI values obtained for crayfish meat were lower than for the meat of several other species, e.g., AI 0.6–0.84 in beef [84], AI 0.56–0.6; TI 1.35–1.50 in lamb [90], AI 0.38–0.39; TI 0.75–0.80 in chicken [53] and AI 0.36–0.37; TI 0.66–0.74 in goose [88].

### 3.3. Culinary Properties of Spiny-Cheek Crayfish Meat

When analysing culinary quality parameters, it can be noted that the abdomen meat has a fine texture and a pleasant smell. Hence, this raw material can be attractive to older people who have a problem with chewing food. The meat is soft, cohesive, springy and juicy, easily chewed, without clearly perceptible connective tissue and fattiness (Table 7).

Compared with the meat of other animals, that of spiny-cheek crayfish has a lower hardness than that of cattle and goat, lower fattiness than that of cattle, goat, sheep, pig, roe-deer and reindeer, and higher juiciness than that of goat and sheep [99]. Due to the high cohesiveness, crayfish meat is not as tender as that of most species of livestock and game. The results of the sensory assessment of crayfish meat texture were confirmed by instrumental analysis (Table 8). However, it is difficult to compare these results with texture analyses of other meat species because of the large variation of parameters used in tests (type of force, extent of deformation, size of shaft, test speed). For example, with the compression plate used in our study, the hardness of crayfish meat measured was 31 N. In studies by other authors, in which penetration shafts were used, hardness was 4.8 N in common carp [100], 25.2 N in common pheasant (*Phasianus colchicus*) [101], 27.7 N in roe deer (*Capreolus preolus*), and 47 N in wild boar (*Sus scrofa*) [102]. A much greater hardness was shown for pig (56 to 100 N) [103,104] and cattle meat (30.5 to 68.7 N) [105]. In turn, Hamre et al. [106], using a compression test at lower deformation values than in our study (60% vs. 80%), reported the hardness of Atlantic salmon fillets at 5.8 N.

Textural properties are also inherently related to the pH of meat. The meat of the spiny-cheek crayfish abdomen had a pH of 7.17, which is similar to the pH of *P. clarkii* abdomen meat (6.98) [16]. However, the observed pH of crayfish meat is much higher than the final pH of livestock meat (5.1–6.2), which causes a lower microbiological stability of crayfish meat despite the decrease in meat pH during storage [107]. A positive feature of spiny-cheek crayfish meat is its light colour and odour profile. In terms of colour, crayfish meat resembles more fish and chicken meat than that of livestock mammals. The high lightness (L*) and low redness (a*) obtained for crayfish meat were comparable with carp [100] and poultry meat [108]. The yellowness (b*) of crayfish meat was low and similar to that of pork [109,110] but much lower than that of beef [111,112], poultry [108], wild boar [113], and carp [100]. The chromaticity of crayfish meat (C) was lower than that of livestock [108,109,111,112] and carp [100] meat.

The fine texture of crayfish meat is due to the small size of muscle fibres and thin connective tissue surrounding the fibres (Table 9). The size of muscle fibres in *F. limosus* muscles, characterised by the CSA, girth, V and H diameters, is smaller compared with that of other aquatic and terrestrial animals. For example, the CSA of muscle fibres in *F. limosus* (85.04 μm^2^) is four times smaller than that in horseshoe crab, *Limulus polyphemus*—484 μm^2^ [114]. Even greater differences in fibre size were observed when comparing crayfish meat to that of various fish species—sea bass [115], common carp [100], tench [116], as well as livestock—pork [103,104], beef [117], chicken [118], and duck [119].

Some of the most important sensory properties of the meat of aquatic animals are the intensive odour and flavour (fishy, seaweed), which depend particularly on storage conditions [120]. In the case of crayfish meat, the detectability of odour and taste that is typical of the animal species is lower than that of beef, goat, reindeer, and roe deer [99]. No sour and metallic flavours, which are detectable in the meat of livestock and game animals, were observed in crayfish meat. The bitter flavour found in crayfish meat was much less intense than that found in the meat of cattle, reindeer, roe deer, goat, rabbit, and even chicken [99]. Spiny-cheek crayfish meat was characterised by a low intensity of odour and taste indices, which are preferable by consumers. For the above reasons, crayfish meat can be an attractive source of protein, fatty acids, and minerals in children’s diet. This group of consumers is particularly demanding as they are reluctant to eat fish meat, a rich source of nutrients (e.g., amino acids, EPA, DHA, etc.) for their developing bodies because of its specific odour and the presence of fish bones.

## 4. Conclusions

The meat of spiny-cheek crayfish is regarded by the food industry as an alternative source of raw material. Our study provides detailed and multidisciplinary results showing the following. (i) Despite the low unit yield of meat extracted from chelae and abdomen, both types of raw material have a high level of protein and a low content of fat. (ii) Chemical scores (CS) and the essential amino acid index (EAAI) calculated for the abdomen and chelae meat of *F. limosus* have a better nutritional protein quality in comparison with the FAO/WHO/UNU (2007) standards. (iii) The quality of crude fat expressed, e.g., by the ratio of n-3 to n-6 fatty acids and three indexes (h/H, IA, IT), is unquestionably beneficial for human health. (iv) Abdomen and chelae meat are both highly nutritious and toxicologically safe concerning heavy metals. (v) The culinary properties of spiny-cheek crayfish meat assessed using numerous parameters of structure, texture, colour and sensory parameters are high for both technologists and consumers. Our study covered a significant gap in the knowledge required by the food industry to begin the exploitation and efficient processing of spiny-cheek crayfish meat. However, ongoing exploitation possibly will eventually lead to the successful eradication of the invasive species, and the food industry must close or may import the material from other countries where *F. limosus* is present as well as change its target species. The results of our research can also be used by chefs. On their basis, it is possible to prepare new, sensory-attractive and nutritious dishes made of spiny-cheek crayfish meat as well as diversify and improve the nutritional value of the already offered dishes made of traditional raw materials.

## Figures and Tables

**Figure 1 animals-11-00059-f001:**
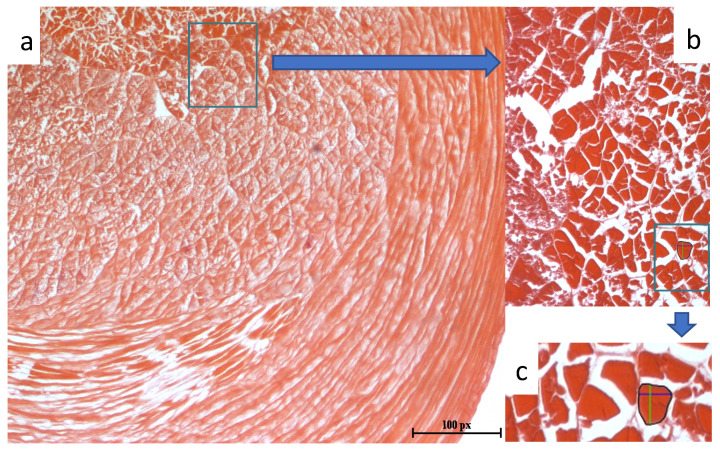
Cross-section of the abdomen meat (**a**), structure elements of muscles (**b**) and fibre cross-section area, fibre girth (black line), horizontal (blue line) and vertical (green line) diameter (**c**).

**Table 1 animals-11-00059-t001:** Yield of meat and hard parts in the abdomen and chelae of *F. limosus.*

Yield	Body Part		Significance of Influence
	Abdomen	Chelae	
Yield of hard parts (% of body part)	62.79 ^a^ ± 0.301	87.07 ^b^ ± 9.922	*
Yield of meat (% of body part)	37.21 ^a^ ± 1.899	12.93 ^b^ ± 3.962	**
Yield of meat (% of body)	8.67 ^a^ ± 0.844	2.69 ^b^ ± 1.328	**

Values are expressed as mean ± standard deviation (SD), ^a,b^—values in rows with different index differ significantly (*p* ≤ 0.05), significance of influence: * *p* ≤ 0.05; ** *p* ≤ 0.01.

**Table 2 animals-11-00059-t002:** Proximate composition of abdomen and chelae muscles of *F. limosus.*

Component (% of Wet Weight)	Abdomen	Chelae	Significance of Influence
Crude protein (%)	18.23 ± 0.732	18.83 ± 1.476	n.s.
Crude fat (%)	0.26 ± 0.024	0.26 ± 0.033	n.s.
Ash (%)	1.59 ± 0.357	2.00 ± 0.810	n.s.
Dry matter (%)	20.28 ± 1.167	21.66 ± 1.755	n.s.
Energy value (kcal/100 g) (kJ/100 g)	75.27 ± 3.102 319.59 ± 13.159	77.52 ± 6.057 329.14 ± 25.721	n.s. n.s.

Values are expressed as mean ± standard deviation (SD), significance of influence: n.s.—non-significant (*p* ≤ 0.05).

**Table 3 animals-11-00059-t003:** Content of elements in the abdomen and chelae meat of *F. limosus*, and percentage of elements covered with 100 g of abdomen and chelae meat.

Elements	Content of Elements		Significance of Influence	% of Elements Covered with 100 g of Meat		Dietary Recommendations UL/AI ^A^/PRI ^B^/MAL ^‡^	References
	Abdomen	Chelae		Abdomen	Chelae		
Macroelements							
Ca (mg kg^−1^)	271.4 ^a^ ± 88.98	13701 ^b^ ± 4951	**	1.09	54.80	2500 mg day^−1^	[56]
K (mg kg^−1^)	4111 ^a^ ± 368.3	3460 ^a^ ± 166.0	**	11.75	9.89	3500 mg day^−1 A^	[57]
Mg (mg kg^−1^)	323.4 ^a^ ± 36.662	405.4 ^b^ ± 32.545	**	9.24–10.78	11.58–13.51	♂350/♀300 mg day^−1 A^	[58]
Na (mg kg^−1^)	890 ^a^ ± 139.911	1319 ^b^ ± 184.59	**	5.93	8.79	1500 mg day^−1 A^	[59]
P (mg kg^−1^)	2746 ^a^ ± 150.23	2482 ^a^ ± 96.86	**	49.93	45.13	550 mg day^−1 A^	[60]
Essential elements							
Al (mg kg^−1^)	25.17 ^a^ ± 3.915	34.69 ^b^ ± 4.654	*	2.57	3.54	1.4 mg bw^−1^ day^−1^/98 mg day^−1 †^/	[61]
Cu (mg kg^−1^)	3.01 ^a^ ± 0.258	3.55 ^b^ ± 0.265	*	6.02	7.10	5.0 mg day^−1^	[56]
Fe (mg kg^−1^)	3.76 ^a^ ± 0.778	5.37 ^b^ ± 0.734	**	4.18–3.76 (2.51)	5.97–5.37 (3.58)	9–10 mg day^−1 B^ (15 mg day^−1 B,C^)	[62]
Zn (mg kg^−1^)	13.66 ^a^ ± 2.131	82.96 ^b^ ± 14.661	**	17.08	103.7	25 mg day^−1^	[56]
Nonessential elements							
Cd (µg kg^−1^)	16.54 ^a^ ± 0.602	13.20 ^b^ ± 1.374	*	0.33	0.26	0.5 mg kg^−1^ ww ^‡^	[63]
Pb (µg kg^−1^)	93.42 ^a^ ± 13.235	69.78 ^b^ ± 5.636	*	1.67	1.40	0.5 mg kg^−1^ ww ^‡^	

Values of the element content are expressed as mean ± standard deviation (SD), ^a,b^—values in rows with different index differ significantly (*p* ≤ 0.05), significance of influence: * *p* ≤ 0.05; ** *p* ≤ 0.01. UL—tolerable upper intake levels, ^A^ AI—adequate intake, ^B^ PRI—population reference intake, ^C^ for menstruating women, ^‡^ MAL—maximum allowed levels, ^†^ calculations were performed for a person with a bw of 70 kg, ww—wet weight of the crustacean meat, bw—person body weight.

**Table 4 animals-11-00059-t004:** Content of essential and nonessential amino acids in the abdomen and chelae muscles of *F. limosus.*

Amino Acids (mg g^−1^ of Meat) (g 100 g^−1^ of Protein)	Abdomen	Chelae	Influence of Body Part
Histidine	3.78 ± 0.304	4.67 ± 0.672	n.s.
	(2.08 ± 0.168)	(2.21 ± 0.198)	(n.s.)
Arginine	19.42 ^a^ ± 2.037	14.93 ^b^ ± 0.306	*
	(10.67 ^a^ ± 1.154)	(7.69 ^b^ ± 0.141)	(*)
Isoleucine	7.42 ± 0.580	8.30 ± 1.104	n.s.
	(4.07 ± 0.265)	(3.89 ± 0.079)	(n.s.)
Leucine	13.12 ± 0.826	15.08 ± 1.898	n.s.
	(7.20 ± 0.368)	(6.92 ± 0.312)	(n.s.)
Lysine	14.10 ± 1.120	15.22 ± 2.010	n.s.
	(7.74 ± 0.628)	(7.28 ± 0.490)	(n.s.)
Methionine	4.54 ± 0.811	5.14 ± 1.256	n.s.
	(2.49 ± 0.432)	(2.45 ± 1.004)	(n.s.)
Tryptophan	5.71 ± 1.538	6.04 ± 3.435	n.s.
	(3.14 ± 0.847)	(3.20 ± 1.441)	(n.s.)
Tyrosine	5.96 ± 0.396	6.45 ± 0.774	n.s.
	(3.27 ± 0.190)	(3.07 ± 0.089)	(n.s.)
Phenylalanine	6.81 ± 0.452	7.1 ± 0.888	n.s.
	(3.74 ± 0.216)	(3.41 ± 0.221)	(n.s.)
Valine	7.28 ± 0.585	7.84 ± 0.952	n.s.
	(3.99 ± 0.263)	(3.72 ± 0.0.132)	(n.s.)
Threonine	6.63 ± 0.422	8.02 ± 1.212	n.s.
	(3.64 ± 0.221)	(3.74 ± 0.093)	(n.s.)
Total essential amino acids (EAA)	94.79 ± 6.956	100.28 ± 11.244	n.s.
	(52.03 ± 3.862)	(47.580 ± 0.306)	(n.s.)
Alanine	9.38 ^a^ ± 0.689	10.85 ^b^ ± 0.961	*
	(5.14 ^a^ ± 0.354)	(5.15 ^b^ ± 0.071)	(*)
Aspartic acid	17.70 ± 1.189	17.85 ± 1.966	n.s.
	(9.71 ± 0.577)	(8.55 ± 0.381)	(n.s.)
Cysteine	2.08 ± 0.749	2.47 ± 1.244	n.s.
	(1.14 ± 0.409)	(0.99 ± 0.436)	(n.s.)
Glutamic acid	29.55 ± 2.139	32.92 ± 4.322	n.s.
	(16.21 ± 0.980)	(15.49 ± 0.110)	(n.s.)
Glycine	12.02 ^a^ ± 1.109	9.38 ^b^ ± 1.290	*
	(6.61 ^a^ ± 0.747)	(4.58 ^b^ ± 0.798)	(*)
Proline	5.22 ^a^ ± 0.263	6.59 ^b^ ± 1.045	*
	(2.86 ^a^ ± 0.092)	(2.94 ^b^ ± 0.075)	(*)
Serine	6.88 ± 0.318	7.26 ± 0.768	n.s.
	(3.78 ± 0.161)	(3.49 ± 0.025)	(n.s.)
Total nonessential amino acids (NEAA)	82.83 ± 5.366	87.32 ± 10.469	n.s.
	(45.47 ± 2.844)	(41.19 ± 0.433)	(n.s.)
EAA:NEAA	1.14 ± 0.0116	1.15 ± 0.0146	n.s.

Values are expressed as mean ± standard deviation (SD), ^ab^—values in rows with different index differ significantly (*p* ≤ 0.05), significance of influence: n.s.—non-significant, * *p* ≤ 0.05.

**Table 5 animals-11-00059-t005:** Nutritional quality of protein in the abdomen and chelae muscles of *F. limosus.*

Amino Acids	FAO/WHO/UNU [25] Scoring Pattern (g 100 g^−1^ of Protein)	CS (% of Scoring Pattern)		
		Abdomen	Chelae	Influence of Body Part
Histidine	1.5	138.53 ± 11.167	147.41 ± 13.208	n.s.
Isoleucine	3.0	135.77 ± 8.828	129.63 ± 2.639	n.s.
Leucine	5.9	122.06 ± 6.243	117.22 ± 5.284	n.s.
Lysine	4.5	172.04 ± 13.961	161.86 ± 10.890	n.s.
SAA	2.2	165.15 ± 38.037	156.43 ± 83.732	n.s.
Phe + Tyr	3.8	184.34 ± 10.639	170.58 ± 8.144	n.s.
Threonine	2.3	158.23 ± 9.599	162.56 ± 4.043	n.s.
Tryptophan	0.6	522.77 ± 141.237	534.17 ± 106.759	n.s.
Valine	3.9	102.42 ± 6.737	95.30 ± 3.383	n.s.
EAAI		151.53 ± 12.434	145.94 ± 9.737	n.s.

Values are expressed as mean ± standard deviation (SD), significance of influence: n.s.—non-significant; SAA—sulfur amino acids, EAAI—essential amino acids index.

**Table 6 animals-11-00059-t006:** Content of fatty acids in the abdomen meat of *F. limosus.*

Fatty Acid (g 100 g^−1^ of Meat)	Mean Value	SD
(C12:0) Lauric acid	0.0006	0.00050
(C14:0) Myristic acid	0.0026	0.00099
(C15:0) Pentadecanoic acid	0.0029	0.00099
(C16:0) Palmitic acid	0.0654	0.02500
(C17:0) Heptadecanoic acid	0.0042	0.00106
(C18:0) Stearic acid	0.0309	0.00891
(C20:0) Arachidic acid	0.0016	0.00113
(C21:0) Heneicosanoic acid	0.0004	0.00042
(C22:0) Behenic acid	0.0022	0.00177
Total SFA	0.1111	0.04144
(C16:1n7) Palmitoleic acid	0.0141	0.00856
(C17:1n7) cis-10-Heptadecenoic acid	0.0036	0.00148
(C18:1n9t + C18:1n9c) Elaidic acid + Oleic acid	0.0812	0.03147
(C20:1n5) cis-11-Eicosenoic acid	0.0010	0.00035
(C20:1n9) cis-9-Eicosenoic acid	0.0038	0.00156
Total MUFA	0.1036	0.04363
(C18:2n6t) Linolelaidic acid	0.0012	0.00057
(C18:2n6c) Linoleic acid	0.0238	0.01068
(C18:3n3) alfa-Linolenic acid [ALA]	0.0087	0.00354
(C20:2n6) cis-11.14-Eicosadienoic acid	0.0102	0.00474
(C20:3n6) cis-8.11.14-Eicosatrienoic acid [DGLA]	0.0010	0.00057
(C20:4n6) Arachidonic acid [AA]	0.0436	0.02319
(C20:3n3) cis-11.14.17-Eicosatrienoic acid [ETE]	0.0025	0.00141
(C20:5n3) cis-5.8.11.14.17-Eicosappentaenoic acid [EPA]	0.0670	0.04179
(C22:6n3) cis-4.7.10.13.16.19-Docosahexaenoic acid [DHA]	0.0018	0.00120
Total PUFA	0.1712	0.09362
Total n-3 PUFA	0.0895	0.0528
EPA + DHA	0.0688	0.04299
Total n-6 PUFA	0.0890	0.03974
Total n-9 PUFA	0.0850	0.03302
PUFA: SFA	1.45	0.2880
n-3/n-6	1.08	0.1235
h:H	3.30	0.4154
AI	0.29	0.0360
TI	0.29	0.0654

Values are expressed as mean ± standard deviation (SD).

**Table 7 animals-11-00059-t007:** Sensory properties of the *F. limosus* abdomen meat.

Parameters (pt.)	Mean Values	Standard Deviation (SD)
Texture:		
Springiness	3.19	0.2394
Cohesiveness	3.23	0.4270
Hardness	2.62	0.6614
Tenderness	1.73	0.2668
Moisture	1.58	0.3536
Juiciness	2.04	0.2205
Perceptibility of connective tissue	1.29	0.2500
Chewiness	1.96	0.2205
Fattiness	1.02	0.0417
Astringency	1.54	0.4167
Odour:		
Boiled meat	1.79	0.5713
Fishy	1.79	0.5833
Seaweed	1.54	0.7120
Taste:		
Boiled meat	1.29	0.2764
Fish	1.38	0.5951
Seaweed	1.38	0.5951
Sweet	1.56	0.5543
Bitter	1.64	0.6138

Values are expressed as mean ± standard deviation (SD).

**Table 8 animals-11-00059-t008:** Texture parameters and physicochemical properties of the *F. limosus* abdomen meat.

Parameters	Mean Values	Standard Deviation (SD)
Hardness (N)	31.03	8.795
Cohesiveness (-)	0.216	0.0340
Springiness (cm)	0.19	0.039
Chewiness (N × cm)	1.20	0.228
pH	7.17	0.180
L*	46.77	2.353
a*	3.07	0.419
b*	6.56	1.847
WI	46.26	2.523
C	7.30	1.578

Values are expressed as mean ± standard deviation (SD). L* (lightness), a* (redness), b* (yellowness), WI (whiteness index), C (chromaticity).

**Table 9 animals-11-00059-t009:** The structural elements of the *F. limosus* abdomen meat.

Parameters	Body Part		Significance of Influence
	Abdomen	Chelae	
Fibre cross-section area (µm^2^)	85.04 ^a^ ± 12.222	38.52 ^b^ ± 7.946	**
Fibre girth (µm)	51.09 ^a^ ± 4.829	32.50 ^b^ ± 2.730	**
Horizontal fibre diameter (H), (µm)	10.89 ^a^ ± 1.348	6.74 ^b^ ± 0.648	**
Vertical fibre diameter (V), (µm)	14.32 ^a^ ± 1.659	9.20 ^b^ ± 0.865	**
Fibre shape (H:V)	0.768 ± 0.0117	0.748 ± 0.0305	n.s.
Endomysium thickness (µm)	1.20 ± 0.1609	1.05 ± 0.0332	n.s.

Values are expressed as mean ± standard deviation (SD), ^ab^—values in rows with different index differ significantly (*p* ≤ 0.05), significance of influence: n.s.—non-significant, ** *p* ≤ 0.01.

## Data Availability

The data presented in this study are available on request from the corresponding author. The data are not publicly available due to privacy.

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
