# Peer review of "Spiny-Cheek Crayfish, Faxonius limosus (Rafinesque, 1817), as an Alternative Food Source"

_animals, 2020, doi:10.3390/ani11010059_

Round 1

Reviewer 1 Report

Would be really great if you could provide some pictures and show different sections in the picture. 

Also, as a possible future activity, would be great if you could bring a statement about how these results would help chefs to develop novel products. 

Author Response

Dear Reviewer,

I would like to thank all reviewers for reading our manuscript and reviewing it. We want to extend our appreciation for taking the time and effort necessary to provide such insightful guidance. The manuscript has been carefully revised in response to your comments and suggestions. Detailed responses to your comments are below. Reviewer’s specific comments start from dashes (-), while our replies follow each comment/suggestion.

Reviewer 1

-Would be really great if you could provide some pictures and show different sections in the picture. 

Following suggestion of the Reviewer 1 authors have added Fig. 1 showing sections of the limosus muscle tissue.

-Also, as a possible future activity, would be great if you could bring a statement about how these results would help chefs to develop novel products. 

Thank you very much for this interesting comment based on which authors has added sentences which inform chefs about the cooking opportunities given by the meat of the spiny-cheek crayfish.

Reviewer 2 Report

The manuscript “Spiny-cheek crayfish, Faxonius limosus (Rafinesque, 3 1817), as an alternative food source” is difficult to follow due to the grammar mistakes and non-sense/incomplete sentences, starting from the abstract. The manuscript needs major revisions.

Major comments to the authors:

  • I suggest authors rewrite the text completely and have a professional translator to revise it.
  • Materials and methods have to be improved, mainly paragraphs 2.2 and 2.3.
  • In line 93 the authors report “Additionally, the yield of meat from the abdomen and chelipeds was assessed”. How?
  • The authors report that fatty acid profiles were determined on n = 50 samples, what about the other chemicals analyzed?
  • The methods used to determine crude protein and crude lipid have to be better described.
  • In Tables 1there are alphabetic letters (a and b), but there is no explanation of what they refer to.
  • The authors used hematoxylin/eosin to study muscle characteristics. However, no images and no description of how they define “CSE, H, and V are reported.
  • The results and discussion are not very clear due to the English language

Author Response

Dear Reviewer,

I would like to thank all reviewers for reading our manuscript and reviewing it. We want to extend our appreciation for taking the time and effort necessary to provide such insightful guidance. The manuscript has been carefully revised in response to your comments and suggestions. Detailed responses to your comments are below. Reviewer’s specific comments start from dashes (-), while our replies follow each comment/suggestion.

Reviewer 2

The manuscript “Spiny-cheek crayfish, Faxonius limosus (Rafinesque, 3 1817), as an alternative food source” is difficult to follow due to the grammar mistakes and non-sense/incomplete sentences, starting from the abstract. The manuscript needs major revisions.

Major comments to the authors:

- I suggest authors rewrite the text completely and have a professional translator to revise it.

Text of the manuscript has been revised by the professional translator to increase readability and remove language issues.

- Materials and methods have to be improved, mainly paragraphs 2.2 and 2.3.

Description of the materials and methods provided in the paragraphs 2.2 and 2.3 were updated to provide more details on e.g. sample digestion and calibration curve. Please see the updated subchapters in the revised manuscript.

- In line 93 the authors report “Additionally, the yield of meat from the abdomen and chelipeds was assessed”. How?

Information on the dissection and calculation steps for the yields were provided in the manuscript (Subchapter 2.1)

- The authors report that fatty acid profiles were determined on n = 50 samples, what about the other chemicals analyzed?

Each subchapter of the Material and methods include information on the number of samples that were used for analysis.

- The methods used to determine crude protein and crude lipid have to be better described.

Description of analytical determination of crude protein and crude lipid was improved with details on the digestion procedures (profiles) and the reagents.

- In Tables 1 there are alphabetic letters (a and b), but there is no explanation of what they refer to.

Explanations for alphabetic letters (a and b) were provided in the caption for the Table 1

- The authors used hematoxylin/eosin to study muscle characteristics. However, no images and no description of how they define “CSE, H, and V are reported.

Information on the tool used to define CSA, H and V were added to the sun-chapter 2.4. Moreover, Fig. 1 showing structure of limosus muscle tissue was added to enrich overall content of the manuscript.

- The results and discussion are not very clear due to the English language

Text of the results and discussion has been revised by the professional translator to increase readability and remove language issues.

Reviewer 3 Report

This is a very thorough study of spiny-cheek crayfish.  The manuscript is well written and it’s easy to access specific data that may be of interest to end users.  It provides all the data that a food processor would need to make an informed decision on the potential for commercial utilization of this species.

The manuscript I reviewed is a standard compositional analysis of a potential commercial seafood. This paper adds to the usual proximate and nutritional analysis a description of color and textural attributes. The paper becomes part of the knowledge base of compositional analysis that is useful to both academics and for industrial applications. It is well written and needs little editing, perhaps a review of English usage although I did not see any glaring problems.

Author Response

Dear Reviewer,

I would like to thank all reviewers for reading our manuscript and reviewing it. We want to extend our appreciation for taking the time and effort necessary to provide such insightful guidance. The manuscript has been carefully revised in response to your comments and suggestions. Detailed responses to your comments are below. Reviewer’s specific comments start from dashes (-), while our replies follow each comment/suggestion.

Reviewer 3

This is a very thorough study of spiny-cheek crayfish.  The manuscript is well written and it’s easy to access specific data that may be of interest to end users.  It provides all the data that a food processor would need to make an informed decision on the potential for commercial utilization of this species.

- The manuscript I reviewed is a standard compositional analysis of a potential commercial seafood. This paper adds to the usual proximate and nutritional analysis a description of color and textural attributes. The paper becomes part of the knowledge base of compositional analysis that is useful to both academics and for industrial applications. It is well written and needs little editing, perhaps a review of English usage although I did not see any glaring problems.

Thank you for your comments. Text of the manuscript has been revised by the professional translator to increase readability and remove language issues.

Round 2

Reviewer 2 Report

I am satisfied with the corrections made. I have no further requests 

Author Response

Thank you very much for all your highly valuable comments. 

Kind regards,

Authors